# ~~Climate~~Weather and resource information as tools for dealing with farmer-pastoralist conflicts in the Sahel

Ole Mertz[1], Kjeld Rasmussen[1], Laura Vang Rasmussen[1,2]

[1]Department of Geosciences and Natural Resource Management, University of Copenhagen, 1350 Copenhagen K, Denmark
[2]International Forestry Resources and Institutions (IFRI), School of Natural Resources & Environment, University of Michigan, Ann Arbor, USA

*Correspondence to*: Ole Mertz (om@ign.ku.dk)

**Abstract.** Conflicts between pastoralists and farmers in the Sahel mainly arise from competition over land and water resources or because of livestock damages to crops. Rather than being linked to larger environmental change processes such
as climate change, ~~they~~conflicts are often caused by inappropriate zoning of land, governance and unequal power relations between stakeholders ~~in the~~. However, conflicts~~.~~ may be affected by more short-term weather and resource information that guide mobility of pastoralists. In this paper, we therefore explore if improved weather and resource information and improvement in its communication could prevent conflicts or reduce their severity. Based on a survey of key stakeholders involved in dissemination of weather and resource information and studies on pastoral access to and use of information, we
conclude that improved information may both reduce and increase the level of conflict, depending on the context. Communication of ~~improved~~ information will need to go beyond just the weather and resource information and also include the multiple options for herd movements as well as providing information on herd crowding and potential conflict areas.

## 1 Introduction

The history of conflicts involving pastoralists and farmers in the West-African Sahel is long~~-~~ (Thébaud and Batterbury, 2001;
Turner, 2004). Conflicts arise from agricultural encroachment on land and pastures traditionally used by pastoralists, or are associated with livestock damages on crops in rain-fed fields and in irrigated gardens. Moreover, there are also classic conflicts between pastoralists on the access to and use of pastures and watering points (wells, boreholes). Especially in the dry season, when prices for accessing water can be high, conflicts may intensify. These conflicts often receive more attention than the well-known symbiotic relationships whereby farmers and pastoralists exchange crop residues and manure. In
addition, herders and farmers are very heterogeneous and overlapping categories, both in terms of production systems, social organization and ethnicity. Across the Sahel, livestock is to an increasing extent owned by groups that are not usually considered to be pastoralists, and groups that are traditionally considered to be pastoralists are increasingly becoming involved in farming and other economic activities (Hesse, 2011), adding to the complexity of conflicts (Beeler, 2006).

Numerous more or less simplistic explanations of the farmer-pastoralist conflicts have been suggested in the scientific literature (see overviews of this large literature by Hussein et al., 1999; Thébaud and Batterbury, 2001; Turner, 2004; Turner et al., 2011; Benjaminsen, 2016). In recent years, for example, it has been claimed in various political arenas that climate change has caused – or aggravated – conflicts, due to its alleged negative impacts on resource availability. Closer

examination of these claims have largely caused them to be refuted as the root causes of conflicts are found in socio-political events and conditions such as inadequate land policies and rent-seeking (Benjaminsen et al., 2012; Benjaminsen, 2016). Some studies do show that extreme weather events such as droughts – whether triggered by climate change or not – may exacerbate existing conflicts (Raleigh and Kniveton, 2012), but the fact that climate variability is perceived to have a much lower impact on livestock productivity in areas, where policies of zoning of pastoral lands arewere implemented and

enforced in the mid-2000s, indicates that climate factors are secondary to policy drivers (Mertz et al., 2010). In the study by Mertz et al. (2010), these zoning policies, which typically entail delimitation of areas for pasture and movement of livestock, were mainly seen to have had a positive impact and conflict mitigating effect, but they have been criticized for being too technocratic and top-down oriented with little pastoral involvement in their conception (Hesse and Thébaud, 2006). In Niger, the limited or inadequate implementation of zoning policies has in some cases had counterproductive effects, creating

negative impacts for pastoralists and potentially exacerbated conflicts (Bonnet and Hérault, 2011; Oxby, 2011) as was also experienced in earlier attempts at controlling, for example, wells and boreholes in eastern Niger (Thébaud and Batterbury, 2001).

Whatever the reason behind, farmer-pastoralist and pastoralist-pastoralist conflicts prevail in many areas in the Sahel. The

20 underlying causes may be social, political or economic, but the direct drivers of specific conflicts are mostly a result of competition for concrete land areas, certain types of vegetation and water resources used for both farming and livestock. As all of these resources are influenced by climate variability, one may hypothesize that better information on the state and changes in resources, and on the weather patterns that influence them, would be useful for mitigating conflicts, at least in the short term and even if it only would be treating the 'symptoms' rather than their root causes.

Hardly any attention has, however, been devoted to what role information about climate, weather, and natural resources might play for conflict resolution. This is surprising as both pastoralists and farmers have been shown to act upon the information available to them and are indeed able to understand more complex probabilistic forecasts, including the risks associated with following recommendations on for example sowing dates and length of the rainy season (Ingram et al., 2002;

Roncoli et al., 2009; Rasmussen et al., 2015). A study in Senegal showed, on the other hand, that pastoralists are reluctant to support information sharing about pastures (Kitchell, 2016). Reasons include that pastures become a 'common property' and this may compromise theirpastoralists' priority access to certain areas, potentially creating additional conflict. Yet, this was not found in northern Burkina Faso, where there was a demand for information and criticism was more directed at its value and the forms of communication (Rasmussen et al., 2014; Rasmussen et al., 2015). In any case, when people are faced with

increasing climate variability, their actions and management strategies will most likely differ depending on the level of knowledge gained about the weather and the resource availability. The question remains whether this knowledge will mitigate or exacerbate conflicts when decisions about resource use and mobility are made.

In the present paper, we discuss the possible linkages between small-scale, localized but common resource conflicts and various information dissemination systems. For example, we look into dissemination systems based on information from satellite data, traditional forecasting methods, and seasonal forecast models. The question asked is whether such information systems, apart from being useful as a basis for day-to-day decisions, will tend to lessen or increase competition for resources and thus the potential for conflicts. We use the limited existing literature to assess the role of information and complement

this with a short questionnaire survey among local government and private stakeholders involved in dissemination of climate, weather, and resource information. The latter involved a questionnaire survey was distributed to staff from key dissemination institutions in West Africa, including provincial agricultural and meteorological services and radio stations in Burkina Faso, Mali and Niger. The staff from the government agencies were either regional directors or leaders of so-called 'focal points' for weather and resource information dissemination and the staff from radio stations were either directors or

journalists working on popular dissemination of this information. All participants therefore had comprehensive local knowledge of the areas they worked in the three countries, but they are mainly policy implementers and agents for transmitting knowledge to the actual land users. The survey was conducted during the "Workshop on the dissemination of agro-hydro-climatological information to final users in the project Knowledge Based Climate Adaptation in West Africa (Original French title: Atelier de Diffusion et de Dissémination de l'Information Agro-Hydro-Climatique aux Usagers

Finaux du Projet ACCIC), held in Ouagadougou, Burkina Faso, 3-5 December 2015. A total of 24 participants took part in the survey. Sixteenwere asked to complete the questionnaire and of these sixteen participants responded to theanonymously. The questionnaire that requested information on their knowledge of cases, where weather or resource or climate information had contributed to resolving or aggravating conflicts, and on their opinion on the role of information as a conflict resolution tool, including how this canconflict resolution may take place. In addition to the survey, notes were taken of discussions

during the workshop to capture opinions and more nuanced details in views on the utility of various types of information systems and theirthe potential of different dissemination systems. This highlighted considerable differences in knowledge of pastoral strategies that – not surprisingly – were most well-known among farmers and pastoralistsparticipants from the driest regions, e.g. eastern Niger and northern Burkina Faso.

Before moving to the results of the survey, we start by identifying the information needs of pastoralists – the potential users of weather and resource information – as they have been largely neglected as recipients of resourcesuch information (Rasmussen et al., 2014; Rasmussen et al., 2015). We then discuss implications of the results for farmer-pastoralist conflict resolution and development of appropriate information systems in the Sahel.

**2 Information needs of pastoralists**

Pastoral societies still rely to a large extent on traditional agricultural and livestock production methods even though the sector to an increasing extent has become a supplier of meat to the coastal regions of West Africa, and thus partly commercialized. As pastoralists are becoming sedentary in many parts of the Sahel, such as the Ferlo of northern Senegal, the competition for land and resources in nearby areas gets more pronounced because pastoralists still rely on varying degrees and types of herd mobility (Adriansen and Nielsen, 2005). Ensuring appropriate and efficient mobility of livestock is thus the key element for which pastoralists need information about the state and expected changes in climate, weather, and resources. Rasmussen et al. (2014) discuss the demand for information among pastoralists on the basis of field work in northern Burkina Faso and find that pastoralists seek information that would facilitate more informed decision-making on herd management. These include the location of the herd in order for it to thrive and make the best of current – and expected future – vegetation and water resources as well as information on markets for selling livestock and purchasing feed and veterinary services, especially if there are expectations of insufficient future availability in pastures and water.

The basis upon which these decisions are taken by pastoralists includes experience from the past, pastoralists' own observations, e.g. signs indicating the arrival of the monsoon and information from family members, friends or hired scouts on vegetation and water resources – as well as prices – often conveyed by mobile phone (Rasmussen et al., 2015; Kitchell, 2016). These traditional information systems are now being complemented by satellite-based information on weather and resource availability, but the role of these new technologies – as well as the full potential of mobile phone technologies – in this decision making process and for preventing or resolving conflicts has yet to be fully explored.

**2.1 Information on climate variability and seasonal forecasts**

Weather patterns and climate variability are ~~of course very~~ important for the availability of vegetation and water resources and improvements in this information ~~on this~~ could ~~be~~ potentially be beneficial for pastoralists~~.~~ (Rasmussen et al., 2015; Kitchell, 2016). The long term effects of climate change ~~that will lead~~, which are likely to include increasing temperatures and fewer but more violent rainfall events (Niang et al., 2014), will of course be relevant for the future survival of pastoralism and farming (Lambin et al., 2014), especially if the observed trends in ~~August~~ rainfall anomalies in August, a crucial months for crops and vegetation in general, continue (Mertz et al., 2012; Nicholson, 2013). However, short term seasonal forecasts are more useful for farmers and pastoralists. Since 1998, the Climate Outlook Forum PRESAO (PREvisions Saisonnières en Afrique de l'Ouest) has created seasonal rainfall forecasts (Tarhule and Lamb, 2003; Patt et al., 2007)~~.~~ and although criticized for their lack of reliability (Fraser et al., 2014) significant advances in the understanding of the West African weather systems have paved the way for better forecasts (Polcher et al., 2011). Such forecasts are mostly seen as an input to farmers' choices of which fields to cultivate and which crops or crop varieties to cultivate. Although farmers, as mentioned above, ~~have been shown to~~may use seasonal forecasts rationally, relatively few farmers ~~do so~~actually use them

(Ingram et al., 2002; Ingram et al., 2008; Roncoli et al., 2009; Roudier et al., 2014), probably because of the inaccessibility of the information. The forecasts are therefore mostly used for national planning purposes and early warnings of crop failure. Analogously, pastoralists' use of seasonal forecasts appears very limited in the Sahel (Rasmussen et al., 2014).

## 2.2 Information on vegetation resources

Vegetation information may be provided by field observation or by satellite-based remote sensing. Obviously, pastoralists themselves monitor vegetation resources and share this information, often using mobile phones but this information is limited in spatial extent and completeness. A number of methods for satellite-based monitoring of vegetation productivity in the Sahel have been developed and could be potentially useful for pastoralists. The current standard methodology is based on analysis of time-series of coarse resolution satellite images, mostly from NOAA AVHRR, SPOT Vegetation and MODIS,

using the normalized difference vegetation index (NDVI) as a proxy for vegetation productivity. Mbow et al. (2013) show that NDVI is sensitive to the species composition, limiting its precision for assessment of fodder production. While in cropped areas the summed NDVI is correlated to crop yield and therefore useful in early warning systems of crop failure, outputs from such monitoring systems are of limited value to pastoralists. For pastoralists the end of the rainy season is the most critical period of the year ~~for pastoralists~~ as they must make decisions on herd location, selling of livestock, splitting of

herds etc. based on information on dry season fodder resources. Unfortunately, satellite-based methods for providing information on available non-green fodder resources in near real-time and with the necessary spatial detail are not presently operational and a suitable method for distributing such information would ~~also~~ have to be developed.

## 2.3 Information on water resources

Pastoralists also need real-time information on water availability in day-to-day decisions, especially during the dry season when ponds and lakes progressively dry out and only water from wells and boreholes is available. This can be provided by remote sensing methods that use high resolution satellite images for monitoring the gradual drying out of surface water resources. However, as wells and boreholes are not always operational, especially those that are operated by pumps that require maintenance, information on access to~~,~~ and availability and price of water is therefore also crucial, and this is

~~presently seldom~~rarely collected and broadcasted widely. A 'pastoral decision-support system' would ideally integrate such information, including information on the physical availability and ~~on the~~ management of the water resources.

## 2.4 Other information types: herd location and markets

Herding decisions are not only based on meteorological information and information on ~~weather and~~the availability of resources ~~but~~. Rather, the decisions also involve knowledge of – or expectations of – the competition for these resources

from herds other than your own. Such information is not publicly available and is therefore obtained through informal networks of family and friends, mostly by mobile phones. Moreover, as pastoralism becomes increasingly commercialized,

decisions are to a greater extent guided by economic criteria, e.g. livestock prices and prices on supplementary feed. Such market information is nowadays available to a substantial part of the pastoralists through the same informal networks based on mobile phones that are gaining increasing importance as the key distribution method.

**2.5 Communication of information to pastoralists:  new communication technology**

~~When~~Since satellite-based crop/vegetation monitoring was first introduced in the 1980s, the information, e.g. in the form of maps of NDVI, ~~was~~has been presented to end-users, such as pastoralists, by radio and television broadcast. Obviously, the impact on pastoralist decision making was quite limited~~.~~ (Rasmussen et al., 2014). The main users of previous efforts to disseminate information, such as the Famine Early Warning System Network (FEWS NET), were primarily government agencies and international donors involved in food relief (Boyd et al., 2013). As mobile phones have become widespread in West Africa, information distribution – and especially the speed of distribution – has been transformed~~,~~ and this ~~needs to~~should be ~~included~~accounted for in new strategies for dissemination of weather and resource information, especially for pastoralists who rely on mobile phones more than any other sector in rural West Africa (Rasmussen et al., 2015). While 'smart-phone' technology may provide a promising avenue for delivering spatially detailed information, their use may, however, be limited in the Sahel. ~~Reasons include that the~~This is because presentation of information to pastoralists that are illiterate and do not have full command of national languages will require careful consideration in order to avoid ~~mis-interpretation~~misinterpretation and inequality in access to the information – the pastoralists themselves use almost exclusively oral communication and services that employ voice messages in local languages are therefore by far the most promising (Rasmussen et al., 2015). Moreover, use of 'smart-phones' rather than traditional mobile phones, will demand more frequent charging, which might prove difficult in remote pastoral communities unless its use is supported by technological development of solar-panel based chargers and/or by battery charging becoming a widely available commercial service. The rapid expansion of mobile phone use among pastoralists also provides a basis for systematic crowd-sourcing and feed-back of localized information to the information service providers, e.g. as discussed by Mueller et al (2015). However, there is so far limited experience with this in Africa.

**3. Conflicts and the role of ~~climate~~weather and resource information**

As mentioned above, very few studies have explored whether ~~climate~~weather and resource information can be used as a tool for resolving conflicts or whether indeed better availability of this information may aggravate conflicts.

**3.1 Results of survey with dissemination stakeholders**

The 16 respondents from the workshop on dissemination provided a somewhat diverse picture on the role of information for conflicts. Three respondents were not aware of concrete cases where ~~climate~~weather or resource information had played a

role in conflict resolution or aggravation, but the remaining 13 provided a total of ~~16~~eight combinations of information types and conflict outcomes (Figure 1). Most respondents provided cases, where information resolved conflicts, which may not be so surprising given the role that these agencies play in disseminating this type of information. However, there were exceptions and these were particularly related to information on water and vegetation resources that could lead to

5 aggravation of conflicts. The cases described were quite diverse and, in the words of respondents, included:

"*Biomass and water information to pastoralists will make them move to favorable areas, provoking conflicts both with farmers and other pastoralists. This is caused by lack of areas for free passage of cattle and because of competition for water in wells*";

"*Too early movement of animals both north to south and south to north caused conflicts in transition zones*";

These are thus both cases of correct information that ~~lead~~leads to clashes between farmers and pastoralists as well as among pastoralists since favorable areas had ~~either~~ not been adequately zoned to receive such a large ~~amount~~number of ~~livestock~~pastoralists and ~~cases where~~ wrong information to farmers led to cultivation in areas less suitable for cultivation, but ~~were~~where livestock would graze during the rainy season.

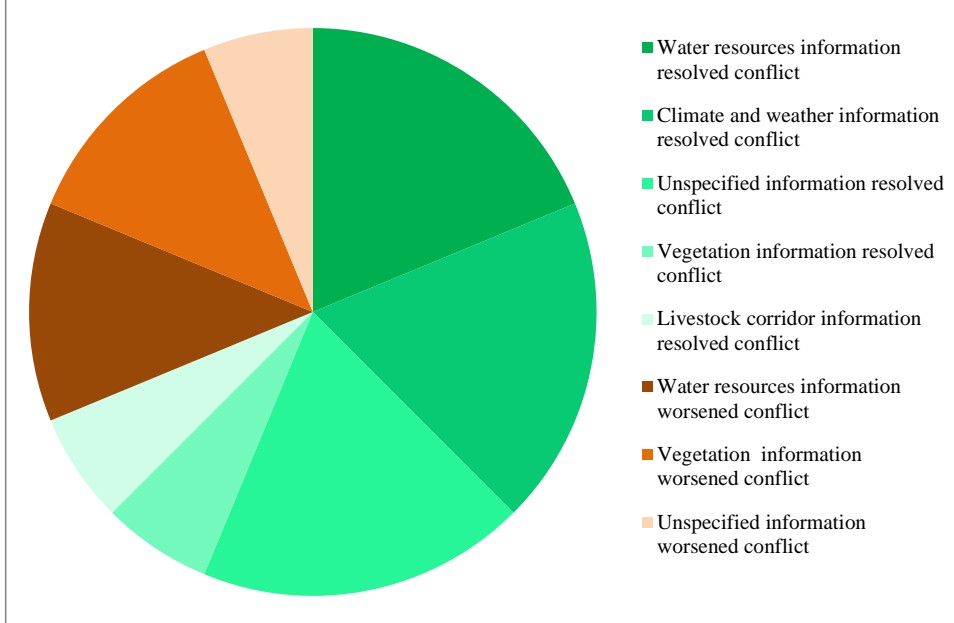

**Figure 1. Distribution of sixteen responses on which types of information may lead to conflict resolution or aggravation as experienced by staff from provincial agricultural and meteorological services and from radio stations in Burkina Faso, Mali and Niger. ~~N=16 (13~~Thirteen respondents provided answers, three provided information on two information categories. Source:**

**questionnaire distributed at "Workshop on the dissemination of agro-hydro-climatological information to final users in the project Knowledge Based Climate Adaptation in West Africa, 3-4 December 2015".**

Interestingly, the survey revealed a new type of conflicts arising from information dissemination: conflicts between farmers and institutions. It was for example expressed that "*Flooding forecasts led farmers to sow on higher and more dry lands and dry spells then caused yields to decline. This caused the farmers to criticize the meteorological department*". Besides lower yields, the expansion into drylands also led to disruption of livestock corridors. This statement highlights the issue of communicating the uncertainty related to the information as uncertain information clearly leaves great room for misunderstanding and miscommunication of risks which can have huge ~~repercussion~~repercussions for pastoralists and farmers' livelihood.

The larger number of responses related to positive impacts of information on conflict resolution was also illustrated by explanations such as:

"*Information given on reduced water level in a dam allowed farmers and pastoralists to agree on the water management and use in the dam*";

"*[Agro-meteorological] information helps pastoralists choose itineraries that avoid newly sown areas by farmers and help farmers avoid planting in livestock corridors*"

"*Information on timing of retracting waters in Lac Chad gives pastoralists the option to avoid islands, where farmers start cultivating*"

Moreover, ~~respondents~~ when asked whether improved information on ~~climate~~weather and resources in certain contexts could assist in conflict resolution, all respondents that provided answers said yes. They illustrated the answers partly with their comments to the previous questions but also elaborated:

"*If information is given so that pastoralists have a variety of options, then they can plan and diversify their movements to avoid all going to the same places. Pastoralists need to have their own information dissemination system improved through proper participation in information system development*"

"*Continued information on and zoning of pasture, livestock corridors, watering areas are needed to avoid further conflicts*"

"*Improvement of the use of mobile phones and other new technologies accessible to pastoralists*"

"*Feed-back to information providers of information needs to be systematic necessary for the systems to get better*"

There was thus a strong emphasis on developing information systems that build on traditional ways of ~~communication~~communicating information and ensure the participation of pastoralists in their conception as well as for

feeding back actual on-site information on resources and weather to improve the information provided. The use of mobile phone technologies was not seen as an obstacle at all as ~~they~~mobile phones have already been appropriated by pastoralists.

There was among representatives from the radio stations a strong and not surprising emphasis on the use of radio transmissions as a way to disseminate information and thus also to contribute to the prevention of conflicts. However, with the exception of Mali, where radio broadcasts were mentioned to have alleviated concrete conflicts, it was not possible to establish whether radio ~~is the best tool to address~~ has been successful in addressing this issue.

### 3.2 Perspectives for ~~climate~~weather and resource information to contribute to resolving conflict

The participants in the workshop all agreed during discussions that there is a need to improve both the quality of information and how it is disseminated as conflicts that could possibly have been avoided, still occur. It is thus evident that farmers and pastoralists in the Sahel make decisions on their use of natural resources on the basis of incomplete information, both about current conditions, e.g. on the spatial distribution of resources, and about probabilities of future events, e.g. the rainfall in the coming rainy season or next year's livestock prices. In this section we therefore discuss the possible consequences of making information on current and future resources more tailored to the needs of pastoralists as a user group, including how it may influence the occurrence of conflicts involving pastoralists.

~~If it would be possible~~A promising option is to produce real-time, spatially explicit information on availability of fodder and water resources (particularly in the dry season) and distribute this to pastoralists, e.g. in graphical form by smart-phone or ~~as voice messages in local languages on an automated phone services as suggested by (Rasmussen et al., 2015). Access to and prices of water are also important for decision making~~ - more appropriately given the limited use so far of smart phones – as voice messages in local languages on an automated phone service as suggested by Rasmussen et al. (2015). This would require investments and partnerships with the private telecommunication sector, but given their success in developing affordable mobile services in Africa, it would appear a feasible proposition. Access to and prices of water are also important for decision making, and information on all these elements would most likely affect decisions concerning location (and possibly splitting) of herds ~~and it~~. This would reduce the probability of making inappropriate short term decisions which might cause increased livestock mortality, economic losses, and conflict with farmers and other pastoralists (Hesse, 2011).

~~In principle, we could thus envision~~Conflicts may arise in a situation where all pastoralists have identical, real-time information about where vegetation and water resources are currently available, and about the access to and price of water resources. As mentioned by the workshop respondents, this may lead many pastoralists to pursue similar strategies, potentially causing increased risk of over-use and subsequent resource depletion and conflict if all descend on the same areas. However, it may also allow pastoralists to obtain information about more options than they otherwise would have had and thus contribute to spreading herds more and ~~thus~~ lowering pressure in each area. The question is thus simply whether a

structured satellite-based information system could provide, if not better, then more information across larger areas than the traditional systems or whether it will just result in more people hearing about a limited number of favorable areas, creating more crowding than previously. Or perhaps the traditional pastoral information systems mentioned earlier are already sufficiently efficient in capturing all available resources and a new system will not make any difference. The only way to answer these questions will of course be to increase the knowledge base on the information-conflict link between farmers and pastoralists. Moreover, attention will also have to be given to potential conflicts between information providers as increased information flows may also result in increased incidences of wrong or inadequate information as in the flood forecasting example listed above. The probabilistic nature of the information needs to be very carefully explained to recipients as too frequent losses of livestock or crops will undermine trust in the system and could escalate into conflicts if damages are severe.

As mentioned in the introduction, it is in any case clear that better information will not be enough to solve conflicts between farmers and pastoralists and among pastoralists. The underlying causes for conflicts are most often related to land policies and (Benjaminsen et al., 2012; Benjaminsen, 2016), and implementation and enforcement of pastoral land zoning are probably some of the best waysis proposed as a way to reduce the number conflicts asif it will clarify land uses for all groups (Mertz et al., 2010; Hesse, 2011). While such land use policies have been implemented in many of the silvo-pastoral zones of the Sahel, they are much less prevalent in the more semi-arid and sub-humid zones dominated by farming and it is often in these areas that conflicts arise when pastoralists search for dry season pastures and water resources. Empowerment of pastoralists and pastoral organizations that allows them to influence land use policies would, among other things, involve assuring equitable access to information and in that sense better climate and resource information could perhaps play a role for increasing the general information level of pastoralists byMoreover, there are unfortunately also many examples of how inadequate or limited implementation of such policies – however well-intended they are – lead to more conflict or other negative outcomes (Thébaud and Batterbury, 2001; Bonnet and Hérault, 2011; Oxby, 2011) Pastoralists and pastoral organizations need to be sufficiently empowered in order to influence land use policies. Top-down technocratic approaches do not facilitate such empowerment (Hesse and Thébaud, 2006) and national and local government will therefore have to truly engage in dialogue with farmers and pastoralists to ensure their involvement and participation. One important first step could be to ensure equitable access to information for both farmers and pastoralists. In particular, more tailored weather and resource information could play a role for increasing the general information level of pastoralists and placing them in a stronger position to argue for their rights to the traditional pastures in predominantly agricultural zones.

## 4. Conclusions

Sahel has for centuries been a scene for fierce competition for land and natural resources, both among pastoralists and between pastoralists and farmers. The great variability in time and space of resource availability requires pastoralists to take

decisions on the basis of incomplete information, sometimes with negative outcomes. Use of modern technologies such as satellite-based earth observation to collect and mobile phones to distribute information on weather, climate variability, vegetation and water resources could be promising for reducing the conflicts that arise over land and access to pasture, and water resources. However, more information may also lead to increased conflict in some cases if it is not managed or communicated in a way that will avoid totoo many herds descending on areas that are too limited in size.

The design of theFuture information systems therefore should not only provideentail actual improvements in access to real-time, spatially explicit weather and resource information. They should also integrate elements such as areas with potential herd crowding and in general be developed with the participation of pastoral communities in order to better target the most pressing needs.

The present paper arrives at these conclusions based on a small survey of stakeholders and a review of the literature. Hence, there is certainly a strong need for studies that take a more systematic look at how weather and resource information intersects with conflict mitigation. Such studies should aim to improve the understanding of the direct linkages between information dissemination, farmer and pastoralist reactions, and the conflict outcomes. Moreover, a set of systematic indicators of successful (or unsuccessful) information dissemination should be developed to make monitoring of the information system performance possible.

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
