# Peer review of "Climate Weather and resource information as tools for dealing with farmer-pastoralist conflicts in the Sahel"

_Earth System Dynamics, 2016_

## Referee Comment (RC1) · C. Hesse (Referee) · 29 May 2016

The paper addresses a very topical and critical issue as farmer-herder conflicts in the Sahel appear to be increasing in frequency and severity – not a week goes by without the media reporting in one or other Sahelian country of clashes over water or access to pastures, or alleged crop damage often with loss of live and livestock. Increasing climate variability as a result of climate change is perceived by policy makers to be a major driver – it is widely believed that the increasingly erratic and variable climate is leading to greater scarcity of resources thus provoking greater competition which inevitably results in violence.

The paper's recognition that while climate change may exacerbate the situation, the underlying factors of herder-farmer conflict reside in the poor resource governance and unequal power relations is a very welcome contribution. Climate variability and uncertainty are what define the Sahel – they are structural characteristics. Local farming and herding communities have institutions and strategies for integrating these characteristics into their production systems, which has always included cooperation such as milk-manure exchanges or water or crop residue and dung exchanges. In the 19th Century, the inner Niger Delta in central Mali under the Dina had a highly complex land use management systems enabling the relatively peaceful co-existence of farming, herding and fishing over the same land area. Sadly this has been progressively dismantled initially by the French when they colonized the country and then more systematically by a series of independent governments. In this sense, the underlying causes of conflict are not competition for resources as such, but the failure or inexistence of appropriate institutions for managing competition and enhancing cooperation – the paper could make this explicit.

The paper quite rightly concludes that improved access to timely and pertinent information, not only on climate and its effect on resource availability but also on other factors that underpin local producers' production strategies and decisions (e.g. market prices), is likely to have a positive effect. The survey, though relatively small, does present some interesting information (e.g. examples of cooperation and better joint management resulting from information on water availability). A presentation of the profile of the respondents and an assessment, if possible, of their understanding of the complexities of the dynamics of Sahelian production systems would help in appreciating how the relevance of the results.

Reference is made in several places to zoning without an explanation of what this entails. Burkina Faso and northern Nigeria have created pastoral "zones" on the assumption that by giving herders their own land, it will protect it from encroachment or alienation and enable more rational production strategies based on sedentarisation and intensification. These have largely failed. Under customary systems, pastoralists

do actively manage the rangelands so as to have wet and dry season pastures, as well as protected areas for weak and young animals, especially during the dry season. This might be considered a form of "zoning", but it is fluid and flexible where access is negotiated and reciprocal. The authors could better explain their understanding of zoning.

Overall, the paper is well-written and well-structured. The title and abstract are informative and true to the paper's content. The authors clearly distinguish their work from others which is clearly credited.

---

## Referee Comment (RC2) · C. Webersik (Referee) · 8 Jun 2016

C. Webersik (Referee)

christian.webersik@uia.no

General comments: The paper is dealing with the role of communication technology for weather forecasting, and its potential in mitigating or increasing the level of conflicts between pastoralists and farmers in the West-African Sahel. The paper addresses relevant questions that fall into the scope of ESD, and more specifically, the special issue of ESD. The paper presents a novel approach in better understanding human-environment interactions, more specifically the role of information on climate variability, weather, and renewable resources may play for conflict resolution. The qualitative methods employed in the paper address the research questions in an adequate way. The overall presentation of the material is well structured and clear, the language is

fluent and precise.

Specific comments: More specifically, there are a few challenges that shall be addressed. In 2.1, the authors argue that seasonal forecasts are not largely used by farmers due to inaccessibility of the information. While this maybe true, it would be good to show a more critical attitude towards seasonal forecasting, questioning whether seasonal forecasts are in fact reliable, especially in the West-African Sahel, where weather stations and recording of weather data are somewhat limited. Hence, it may be advisable to consult the literature on weather forecasting in order to assess the reliability of such data. The reliability aspect of weather forecasting data is also relevant for 3.1, when the authors discuss conflicts between farmers and institutions due to unreliable flooding forecasts.

In 2.4, the authors argue that mobile phones are key in gaining and distributing information. Since smart phones have limited penetration within the population, it seems that "conventional" mobile phones are spreading fast. The authors could be more specific about the use of conventional mobile phones, whether the sharing and accessing of information is mainly voice-based, or whether some information is also accessed via systems that allow the display of web content on conventional mobile phones, such as Opera Mini Mobile Browser or SMS services.

In 3.1, a key aspect of this paper, namely crowd sourcing of weather-related information, deserves more attention. The potential of pastoralists in feeding back information on actual on-site situational data should be explored further. One wonders whether there are examples from other African countries where such crowd-sourcing activities may take place? Given that crowd sourcing is in its infancy for most applications and in the Global North, this maybe not yet the case in relation to weather forecasting.

Technical corrections: Page 4, line 9, delete word "August", and page 9, line 3, author is missing at end of sentence.

---

## Referee Comment (RC3) · Anonymous Referee #3 · 21 Jun 2016

1. Does the paper address relevant scientific questions within the scope of ESD? Yes. The paper can be seen as a reconnection between climate and biophysical data for planning needs and prevention of conflicts between farmers and pastoralist in the Sahel. The paper focuses on the perception and utility of data for local farmers. It is important to re-qualify what conflicts means. The large geopolitical conflicts are related to socio-political conflict, but many micro-conflicts arises in relation to competition for resources that are affected by climate change. 2. Does the paper present novel concepts, ideas, tools, or data? The data and tools are not novel, the outcome of the analysis are very novel and informative. The issue of informed decision making by local stakeholders has been described as suitable way to address various risks and climate

related threats. This paper helps re-qualify the information needs and the subsequent improvement needed to address current challenges. The paper was not explicit about sources of conflicts in the Sahel and reduced those to socio-politically conflicts. Many studies argue that that shortfall in zoning and governance are exacerbated by climate change. Another dimension is political marginalization. Usually pastoralists are not usually part in exercising their duties (no real territories) and claiming their civil right is not part of their culture. Conflicts may also arise because of the discrepancy between free riders citizens and compliant citizen opposing the agriculturalists and the pastoralists. From typical examples in the Sahel, drought are favorable conditions for violence because of resource scarcity (e.g. doi: 10.1177/0022343311427343 ) 3. Are substantial conclusions reached? The conclusions are very useful, yet I let me in doubt how those could derived from a limited sample of practitioners, who we do not know about their origins and who they represent. The paper did not clarify that in many instances, the relationships between pastoralist and farmers are based on traditional rules. Taking the example of Senegal, the symbiotic relationship—farmers had their pastoralists during the dry season to manure the land and harvest the crop residues—was a real social norm. Farmers symbolically prepared for some sort of rewards for the pastoralists staying in their field over the dry season. Every farmer had therefore his pastoralist. This linkages was joint points between two civilizations where each player had a reward. The conflict raised when more fragmentation took place, monetization of the economies, and transformation of social norms imposed by new land management systems. These shift in practices explains a transition to more economic centric goals rather than cultural dimension of such activities. The reality might be more complex than what was deemed important here, that is the importance of a zoning approach. Traditionally, moving herds is part of a civilization, where shepherds are tested for their ability to move around the animals without losing any. As part of the culture, a zoning as a new dimension of management might be useful but not enough to reduce the importance of traditional herd's displacement! 4. Are the scientific methods and assumptions valid and clearly outlined? I have real concern that strong conclusion could

be derived from a sample of 26 participants who were selected in a very unclear way and then use their qualitative assessment to derive quantitative outputs. I am not a socio-economist, but a response to the method approach used will be very enlightening. Additionally, the fine time and spatial scale information requirement is generally true need for community but recent CCAFs experience in Senegal (using rural radios), or METAGRI project in Cote d'Ivoire, clearly demonstrated a general interest on aggregated climate services. The sawing dates and length of the growing season were useful for decision to be made on-farm to adapt to likely seasonal profiles. 5. Are the results sufficient to support the interpretations and conclusions? Intuitively YES, the results and novel lenses of exploring the challenges are useful. I was unclear about the merged approach of identifying information needs for farmers and pastoralist. To me the two groups have different information needs but the analysis seems to put them in the same group when exploring data and information requirements. The study is more like an expert knowledge assessment and will be easily dismantled by control of facts from practitioners in the ground. The authors recommended a real time system of disaggregated data at daily basis to help farmers and pastoralist. Is there any example of such a system in the world? Is that possible given the level of technology we have in Africa? 6. Is the description of experiments and calculations sufficiently complete and precise to allow their reproduction by fellow scientists (traceability of results)? No. The qualitative information might not be appropriate to generate quantitative evidence and the sample size to me very limited. The experts might not have covered the range of opinions needed to develop the outputs. The traditional knowledge is not fully accounted in the modern assessment of information needed for informed decision by herders. Also, a distinction needs to be made between data and information. Information adds more value to data and gives examples of application rather than pure data dissemination. 7. Do the authors give proper credit to related work and clearly indicate their own new/original contribution? Yes. 8. Does the title clearly reflect the contents of the paper? Yes 9. Does the abstract provide a concise and complete summary? Yes. . . But the disregard to climate change trends is quite worrisome to me. Many previous

studies showed the importance of climate change in the Sahel including perception analysis of such trends. The authors themselves are part of that community. 10. Is the overall presentation well-structured and clear? Yes. Except Figure 1. We do not know is that is % or absolute values. In any case I doubt about representativeness of the information given the very small sample size? 11. Is the language fluent and precise? YES 12. Are mathematical formulae, symbols, abbreviations, and units correctly defined and used? N/A 13. Should any parts of the paper (text, formulae, figures, tables) be clarified, reduced, combined, or eliminated? Figure 1 14. Are the number and quality of references appropriate? YES 15. Is the amount and quality of supplementary material appropriate? N/A

---

## Referee Comment (RC4) · Anonymous Referee #4 · 3 Jul 2016

A) This paper explores the role of improved information and communication about weather, climate and natural resources in conflict and conflict resolution between pastoralists and farmers in the Sahel over land and water resources as well as crops and livestock. This is a relevant scientific issue that has yet not been adequately addressed in the literature and falls within the general scope of ESD, in particular this special issue. The analysis is based on a questionnaire and qualitative data from feedback of workshop participants. While the paper offers some valuable results and discussion, some weaknesses are to be addressed.

B) The literature review can be improved, to partly compensate for other deficiencies and increase the substance of the paper. 1. Initial statements on pastoralist-farmer

conflicts need to be better connected to the peer-reviewed literature (e.g. the first two references are working papers). 2. References in the introduction are quite selective (e.g. only citing three papers on the climate-conflict link, despite a body of recent literature). 3. Since the paper aims to enter new ground by connecting different issues, a more systematic review is needed on pastoralist-farmer conflicts related to key resources (land, water, livestock, crops, ..) and why information on those is relevant for farmers and pastoralists in the Sahel, as suggested in the paper. 4. Further it is important to make the meaning of conflict used in this paper more explicit which apparently deals with small-scale conflicts. 5. Some statements deserve better justification, e.g. the authors associate references with "simplistic explanations" but it is not discussed why they are simplistic and which better explanations could be given.

C) Drawing on the literature more specific research questions and hypotheses could be derived, leading to the core part of the paper, the discussion of different categories of information sources, means of communication and relevant data. One strength of the paper is to classify the availability of information sources (satellite data, traditional forecasting, seasonal forecast models, new communication technologies), information content (on weather and climate variability, vegetation and water resources, herd location and markets) and limitations (non-green fodder resources, lack of real-time and spatial details, restrained distribution systems). Interesting is the limited contribution of radio and TV broadcasts for delivering spatially detailed information and the role of mobile phones to informal networks of family and friends providing information to pastoralists even in remote areas. Significant are also considerations on the zoning of land, governance and unequal power relations between stakeholders which would deserve further discussion.

D) While the subject of research is novel, this is not matched by the methodological approach which is based on rather simple tools. A conceptual framework of analysis is missing, and there are no graphs or tables to structure the analysis which would be helpful. Data are based on a questionnaire distributed among participants of a workshop held in Burkina Faso in December 2015, including stakeholders involved in dissemination of climate and resource information in West Africa. In addition, arguments and opinions are drawn from notes of discussions during the workshop to represent views on information systems and their dissemination among farmers and pastoralists. The empirical approach is straightforward but is lacking depth and breadth, relying only on a quite limited data-base. 1. Of 24 workshop participants 13 provided 16 combinations of information types and conflict outcomes (Figure 1) which is not an impressive sample for a quantitative assessment. More information is needed whether Figure 1 presents absolute or relative numbers, on the participants of the workshop and how representative they are, the precise questions asked, the results of the workshop and whether they are published elsewhere. 2. Most respondents referred to cases where information resolved conflicts, while in some cases information on water and vegetation tended to aggravate conflicts. Particularly interesting may be whether some respondents suggested that information both increases and decreases conflict.

E) More interesting than the mere numbers are the qualitative viewpoints of participants on the linkages between information and conflict. Notwithstanding the limited data-base, the results appear novel as they draw possible linkages between weather- and resource-related information and dissemination systems and aggravation or resolution of conflict. 1. However, no theoretical explanations are given whether and when information is leading to competition, sharing or better distribution of resources. Did conflicts emerge because of correct information or due to wrong and lacking of information? This could be identified as a research questions earlier in the paper. 2. The paper emphasizes agreement among the workshop participants "that there is a need to improve both the quality of information and how it is disseminated" (page 8). This raises the questions which indicators could be used to measure improvement and how to improve them. 3. Although representatives from three countries (Burkina Faso, Mali and Niger) were participating, almost no country-specific experiences are presented. Burkina Faso was shortly mentioned twice, Mali once and Niger not at all. Any information on the differences or similarities of these countries would be helpful. 4. The

survey revealed a new type of conflicts between farmers and institutions from information dissemination (page 7). This is an interesting point that could be elaborated further. 5. The paper emphasizes how important traditional ways of information and communication are and to ensure the participation of pastoralists (page 8). Here it would be valuable to include a little more about these traditional ways. 6. Generally it would be useful to have a table on conflicting issues and how information could address them, following the classification mentioned in C). This might include cases where similar information among different groups leads to similar strategies that increase the risk of overuse and depletion of resources, as well as cases where more options are created that reduce conflict and the added value of modern vs. traditional information dissemination. 7. Another point that deserves further discussion is the empowerment of pastoralists by equitable access to information to influence land use policies (page 9). How are empowerment and information related in this context?

F) Despite the methodological limitations, some of the analysis, results and conclusions are interesting and worth publication, with the suggestions given above. The conclusions are too short and unspecific to represent the content of the paper.

---

## Author Comment (AC1) · 21 Jul 2016

Thank you very much for the positive notes on the relevance of the paper, the structure of the text as well as acknowledging the value of the relatively limited survey. The three constructive suggestions for changes can be dealt with relatively easily:

1) Making it more explicit that it is the failure or inexistence of appropriate institutions for managing competition for resources and enhancing cooperation.

a. We agree that this is a crucial element and this can be done by using more of the existing literature on how governments have dealt (or not dealt) with issues such as farmer-herder conflicts.

2) A presentation of the profile of the respondents and an assessment, if possible, of their understanding of the complexities of the dynamics of Sahelian production systems would help in appreciating the relevance of the results.

a. Further presentation of their profiles is no problem, but it would be more daring to assess their understanding of complexities of the Sahelian production systems. However, we do have quotes from the workshop that clearly show strong differences in understandings, e.g. the participants from eastern Niger had a much more pronounced and detailed knowledge of pastoral issues compared to others. We can – with caution – integrate this in the text.

3) A better explanation of how zoning is understood given that government efforts have failed in Burkina Faso and northern Nigeria.

a. We can easily explain the concept of zoning and how it is done differently by governments or by pastoralists themselves. We were of the impression that zoning in northern Burkina Faso had been somewhat successful, but this was information from the late 2000s and might not be relevant anymore. It would be great if the reviewer could direct us to literature on this failure as we are not aware of it.

---

## Author Comment (AC2) · 21 Jul 2016

Thank you very much for the positive notes on the general aspects of the paper. Below we respond to the specific comments and constructive suggestions for changes:

1) It would be good to show a more critical attitude towards seasonal forecasting, questioning whether seasonal forecasts are in fact reliable, especially in the West-African Sahel, where weather stations and recording of weather data are somewhat limited. Hence, it may be advisable to consult the literature on weather forecasting in order to assess the reliability of such data. The reliability aspect of weather forecasting data is also relevant for 3.1, when the authors discuss conflicts between farmers and institutions due to unreliable flooding forecasts.

[Figure]

a. We agree that this is an important issue and we can add some lines on the reliability of forecasts, how it has evolved over time and what it means for the relationship between farmers and institutions. This will of course be based on other sources. We are not sure whether it will be possible to integrate this in 3.1, but it can be a discussion point related to the 'new' conflict between the government institutions disseminating information and users that are affected by information – it may not be a matter of 'wrong' information, but rather about how it has been interpreted along the communication pathway from producer to user.

2) The authors could be more specific about the use of conventional mobile phones, whether the sharing and accessing of information is mainly voice-based, or whether some information is also accessed via systems that allow the display of web content on conventional mobile phones, such as Opera Mini Mobile Browser or SMS services.

a. Phones are mainly used for voice-based information exchange, especially since there is still high illiteracy among pastoralists. We can add a sentence about this.

3) The potential of pastoralists in feeding back information on actual on-site situational data should be explored further. One wonders whether there are examples from other African countries where such crowd-sourcing activities may take place? Given that crowd sourcing is in its infancy for most applications and in the Global North, this maybe not yet the case in relation to weather forecasting.

a. We can integrate the ideas in Müller et al 2015 (http://onlinelibrary.wiley.com/doi/10.1002/joc.4210/full) in this discussion

4) Technical corrections: Page 4, line 9, delete word "August", and page 9, line 3, author is missing at end of sentence.

a. Corrections well noted
* * *

---

## Author Comment (AC3) · 21 Jul 2016

Thank you very much for all the positive notes on the different paper assessment criteria. Here, we focus on responding to the critical comments and constructive suggestions:

1) The paper was not explicit about sources of conflicts in the Sahel and reduced those to socio-politically conflicts. Many studies argue that shortfall in zoning and governance is exacerbated by climate change. Another dimension is political marginalization. Usually pastoralists are not usually part in exercising their duties (no real territories) and claiming their civil right is not part of their culture. Conflicts may also arise because of the discrepancy between free riders citizens and compliant citizen

opposing the agriculturalists and the pastoralists. From typical examples in the Sahel, drought are favorable conditions for violence because of resource scarcity (e.g. doi: 10.1177/0022343311427343 )

a. We agree that extreme climate conditions can be a source of specific conflicts, but as these specific events have always occurred they are difficult to link to climate change. So yes, climate change may exacerbate conflict, but it could also do the opposite. We find that the occurrence of drought is a baseline condition in the Sahel, but we can indeed nuance the debate somewhat in the paper. We also agree that pastoralists are politically marginalized and this is indeed part and parcel of the broader national and international conflicts in the region. We were somewhat hesitant about getting too much into these larger-scale conflicts, but acknowledge that political issues also affect the small-scale conflicts, so we can add some elements to this discussion.

2) The conclusions are very useful, yet I let me in doubt how those could derived from a limited sample of practitioners, who we do not know about their origins and who they represent.

a. We will add more information about the respondents and what they represent as well as their opinions (see also comment by other reviewer)

3) The traditional relationship and exchange culture between farmers and pastoralists is not described.

a. It is correct that we did not go much into this. While we certainly acknowledge its importance, we find that more elaborate discussions would be repeating what is well known from the extensive ethnographic and anthropological literature. We can add some sentences on this related to the type of knowledge we acquired from the respondents, acknowledging that we in this case do not have the direct opinions of the farmers and pastoralists.

4) Concern with the methodology in the paper and the number of respondents:

[Figure]

a. They paper is basically an exploratory exercise that emanated from the fact that this workshop was held and provided the opportunity to gauge opinions from different stakeholders involved in dissemination of information. We certainly appreciate the limitations of this approach and therefore have also used our experience from field work reported in other papers (Rasmussen et al. 2014 and 2015) as well as the wider literature. We can add some methodological reflections in the conclusion as this topic definitely requires more substantial field based research across a range of different stakeholders.

5) The fine time and spatial scale information requirement is generally a true need for communities but recent CCAFs experience in Senegal (using rural radios), or META-GRI project in Cote d'Ivoire, clearly demonstrated a general interest in aggregated climate services. The sowing dates and length of the growing season were useful for decisions to be made on-farm to adapt to likely seasonal profiles

a. We take note of this, but do not really see which changes this should generate in the paper.

6) I was unclear about the merged approach of identifying information needs for farmers and pastoralist. To me the two groups have different information needs but the analysis seems to put them in the same group when exploring data and information requirements.

a. It is not our intention to say that farmers and pastoralists have the same needs – but they can use the same types of information for their specific (and often different) needs. We hope this comes across in the paper, but we will of course double check that it does

7) The study is more like an expert knowledge assessment and will be easily dismantled by control of facts from practitioners on the ground. The authors recommended a real time system of disaggregated data at a daily basis to help farmers and pastoralist. Is there any example of such a system in the world? Is that possible given the level of

technology we have in Africa?

a. The study is actually based on statements from practitioners who work on the ground, so we do not agree that it is not rooted in actual practices. There are many examples of real-time information services in the World and the technology is not complicated. In Africa, where the use of mobile technologies are often more advanced than elsewhere, a combination of political commitment and private investments would of course be needed to implement it. We can make these issues more clear in the paper.

8) The qualitative information might not be appropriate to generate quantitative evidence and the sample size to me is very limited. The experts might not have covered the range of opinions needed to develop the outputs. The traditional knowledge is not fully accounted in the modern assessment of information needed for informed decisions by herders

a. We are well aware that this is not a quantitative and representative sample and although we show a distribution of answers, this is not a quantitative analysis as such, but merely to provide the reader with an overview of responses. The study could, in principle, be replicated by asking the same respondents the same questions (respondents' names are on file, but they are kept anonymous for publication purposes). We will consider whether the figure and its caption can be reworked to not signal a strict quantitative approach, but rather an overview of responses, e.g. in Table form. See also our response to the previous comment.

9) Also, a distinction needs to be made between data and information. Information adds more value to data and gives examples of application rather than pure data dissemination.

a. Thanks, we will check the paper for consistency.

10) But the disregard to climate change trends is quite worrisome to me. Many previous
studies showed the importance of climate change in the Sahel including perception analysis of such trends. The authors themselves are part of that community.

a. First of all, we do mention certain elements that might be linked to climate change, e.g. the crucial nature of August rainfall. Secondly, we have to be very cautious when discussing climate change in the Sahel compared to the inherent climate variability. There is no doubt that the droughts in the 1970s and 1980s represented a (temporary?) shift in climate patterns, but this has been difficult to link to anthropogenically driven climate change. That climate change will have an influence in the future is certain, but for now models at regional scale have great difficulties in agreeing on predictions of climate change for the region. Given the limited number of meteorological stations in the region, it is also very difficult to claim actual changing climate in the past. Hence, we agree with the referees that acknowledge the highly variable climate in the Sahel as a fundamental condition in which resource conflicts exist. In previous work that the referee alludes to, we have shown that climate factors actually have a limited effect on farming and pastoral decisions, probably because the variable climate has always existed.

11) Figure 1. We do not know is that is % or absolute values. In any case I doubt about representativeness of the information given the very small sample size?

a. The pie chart indicates absolute numbers – this will be added to the graph if we keep it. The question of representativeness has been answered above.

---

## Author Comment (AC4) · 21 Jul 2016

Thank you very much for a very detailed review and the positive notes on the relevance. We have responded to all of the constructive suggestions, most of which we can address in the revision:

1) Initial statements on pastoralist-farmer conflicts need to be better connected to the peer-reviewed literature (e.g. the first two references are working papers).

a. It is not entirely correct that they are working papers – they are published as a book chapter and in a series. But point taken that more of the peer-reviewed literature could be cited here

2) References in the introduction are quite selective (e.g. only citing three papers on the climate-conflict link, despite a body of recent literature).

a. We can indeed add more citations to this section, but perhaps the referee could point us to some key references. We thought we had captured some of the most relevant for the current debate. As we focus on small-scale resource conflicts and not the wider climate-conflict links, the latter is mainly used to set the context for the paper.

3) Since the paper aims to enter new ground by connecting different issues, a more systematic review is needed on pastoralist-farmer conflicts related to key resources (land, water, livestock, crops, ..) and why information on those is relevant for farmers and pastoralists in the Sahel, as suggested in the paper.

a. We acknowledge the benefits of doing systematic reviews, but we find that it is beyond the scope of this paper to do a full systematic review of the pastoralist-farmer conflict literature. We believe that we have captured the (few) papers that discuss information needs in relationship to farmer-pastoral conflict but if we have missed some important pieces, we will of course be pleased to integrate these. The paper is essentially an explorative piece of ideas and to implement a systematic review procedure at this point would not be possible. It probably would also not be worth the effort given the limited literature that speaks of climate and resource information for pastoralists as documented in the sources used in the paper (Rasmussen et al. 2014, 2015).

4) Further it is important to make the meaning of conflict used in this paper more explicit which apparently deals with small-scale conflicts.

a. Point taken – this will be corrected. We can clarify that we work with small-scale conflicts that despite their limited individual extent are an issue of relevance across the Sahel.

5) Some statements deserve better justification, e.g. the authors associate references with "simplistic explanations" but it is not discussed why they are simplistic and which

better explanations could be given

a. Agree, we can elaborate on these

6) Drawing on the literature more specific research questions and hypotheses could be derived, leading to the core part of the paper, the discussion of different categories of information sources, means of communication and relevant data

a. We are not entirely sure what is meant here, but we can work on sharpening the research questions if they are not clear enough. We believe that we are indeed doing what is suggested here.

7) Comments related to the survey at the workshop and the figure.

a. We realize that figure 1 does indeed appear as an attempt to make a quantitative assessment of the data, but in reality it was meant to give the reader an overview of the responses. We will consider presenting the figure and caption in a different way or at least make it clear that we know it is not a representative sample that can be the basis for statistical analysis (which is also why we didn't attempt the latter). b. The figure represents absolute numbers c. We have reported most of the statements of interest from the questionnaire and discussions.

8) No theoretical explanations are given whether and when information is leading to competition, sharing or better distribution of resources. Did conflicts emerge because of correct information or due to wrong and lacking of information? This could be identified as a research questions earlier in the paper

a. We do have an example of conflict arising because of 'wrong' information – to quote the respondents. The answer was not elaborated on by the respondents and it could be an issue of how the information was interpreted along the communication pathway rather than actually being wrong information. We will see if more information can be derived from the discussions at the workshop that were also recorded. We do not really see the need to develop this as a full research question.

9) The paper emphasizes agreement among the workshop participants "that there is a need to improve both the quality of information and how it is disseminated" (page 8). This raises the questions which indicators could be used to measure improvement and how to improve them

a. Very relevant point. We do discuss models for improving, but not indicators for measuring their success. We can elaborate on this although it will be somewhat speculative.

10) Although representatives from three countries (Burkina Faso, Mali and Niger) were participating, almost no country-specific experiences are presented. Burkina Faso was shortly mentioned twice, Mali once and Niger not at all. Any information on the differences or similarities of these countries would be helpful

a. Partly because of the limited number of respondents, we decided not to discuss country differences too much. However, based on the discussions there was a clear difference not so much between nationality, but more based on whether they worked in more or less pastoral areas. We can try to elaborate on this latter difference.

11) A new type of conflicts between farmers and institutions from information dissemination (page 7). This is an interesting point that could be elaborated further.

a. Yes, we can add some more comments to this as it is indeed a type of conflict that may arise more if information systems are developed and implemented to a higher extent in the future

12) The paper emphasizes how important traditional ways of information and communication are and to ensure the participation of pastoralists (page 8). Here it would be valuable to include a little more about these traditional ways.

a. These traditions have been documented in detail by Rasmussen et al. (2014, 2015), but we can consider adding a few more details here

13) Generally it would be useful to have a table on conflicting issues and how information could address them, following the classification mentioned in C). This might include cases where similar information among different groups leads to similar strategies that increase the risk of overuse and depletion of resources, as well as cases where more options are created that reduce conflict and the added value of modern vs. traditional information dissemination.

a. Good idea, but we are not entirely sure how to construct such a table without making some unfounded simplifications. We can consider the idea, but are not sure that it will work.

14) Another point that deserves further discussion is the empowerment of pastoralists by equitable access to information to influence land use policies (page 9). How are empowerment and information related in this context?

a. This was also raised by another referee and we will try to add some elements to this discussion.

15) The conclusions are too short and unspecific to represent the content of the paper.

a. Point taken – conclusions will be elaborated (but still kept relatively short).

---

## Author Response (AR2)

**Responses to referees:**

Ced Hesse:

Thank you very much for the positive notes on the relevance of the paper, the structure of the text as well as acknowledging the value of the relatively limited survey. The three constructive suggestions for changes can be dealt with relatively easily:

1) Making it more explicit that it is the failure or inexistence of appropriate institutions for managing competition for resources and enhancing cooperation.
   a. We agree that this is a crucial element and we have used more of the existing literature on how governments have dealt (or not dealt) with issues such as farmer-herder conflicts. There is relatively limited research available on this, however.
2) A presentation of the profile of the respondents and an assessment, if possible, of their understanding of the complexities of the dynamics of Sahelian production systems would help in appreciating the relevance of the results.
   a. We have added some more information on the participant profiles at the end of the introduction. It is more difficult to assess their understanding of complexities of the Sahelian production systems, but we do have quotes from the workshop that clearly show strong differences in understandings, e.g. the participants from eastern Niger had a much more pronounced and detailed knowledge of pastoral issues compared to others. We have – with caution – integrated this in the analysis.
3) A better explanation of how zoning is understood given that government efforts have failed in Burkina Faso and northern Nigeria
   a. We now use some references from Niger, where the zoning and pastoral policies have been analyzed more and have been less successful.

C. Webersik:

Thank you very much for the positive notes on the general aspects of the paper. Below we respond to the specific comments and constructive suggestions for changes:

1) It would be good to show a more critical attitude towards seasonal forecasting, questioning whether seasonal forecasts are in fact reliable, especially in the West-African Sahel, where weather stations and recording of weather data are somewhat limited. Hence, it may be advisable to consult the literature on weather forecasting in order to assess the reliability of such data. The reliability aspect of weather forecasting data is also relevant for 3.1, when the authors discuss conflicts between farmers and institutions due to unreliable flooding forecasts.
   a. We have added a sentence on the reliability in section 2.1
2) The authors could be more specific about the use of conventional mobile phones, whether the sharing and accessing of information is mainly voice-based, or whether some information is also accessed via systems that allow the display of web content on conventional mobile phones, such as Opera Mini Mobile Browser or SMS services.

a. Phones are mainly used for voice-based information exchange, especially since there is still high illiteracy among pastoralists. We have added a sentence about this in section 2.5.

3) The potential of pastoralists in feeding back information on actual on-site situational data should be explored further. One wonders whether there are examples from other African countries where such crowd-sourcing activities may take place? Given that crowd sourcing is in its infancy for most applications and in the Global North, this maybe not yet the case in relation to weather forecasting.

    a. We have integrated the ideas in Müller et al 2015 (http://onlinelibrary.wiley.com/doi/10.1002/joc.4210/full) in this discussion in section 2.5

4) Technical corrections: Page 4, line 9, delete word "August", and page 9, line 3, author is missing at end of sentence.

    a. Corrected

Anonymous referee #3:

Thank you very much for all the positive notes on the different paper assessment criteria. Here, we focus on responding to the critical comments and constructive suggestions:

1) The paper was not explicit about sources of conflicts in the Sahel and reduced those to socio-politically conflicts. Many studies argue that shortfall in zoning and governance is exacerbated by climate change. Another dimension is political marginalization. Usually pastoralists are not usually part in exercising their duties (no real territories) and claiming their civil right is not part of their culture. Conflicts may also arise because of the discrepancy between free riders citizens and compliant citizen opposing the agriculturalists and the pastoralists. From typical examples in the Sahel, drought are favorable conditions for violence because of resource scarcity (e.g. doi: 10.1177/0022343311427343 )

    a. We agree that extreme climate conditions can be a source of specific conflicts, but as these specific events have always occurred they are difficult to link to climate change. So yes, climate change may exacerbate conflict, but it could also do the opposite. We find that the occurrence of drought is a baseline condition in the Sahel, but we have nuanced the debate somewhat in the paper, e.g. in the introduction. Moreover, as the paper deals more with weather information than with climate information, we have changed the title and the wording where relevant. We also agree that pastoralists are politically marginalized and this is indeed part and parcel of the broader national and international conflicts in the region. We did not want to get into these larger-scale conflicts, however – we understand that such political issues also affect the small-scale conflicts, but as we do not have data on it we have not developed this further.

2) The conclusions are very useful, yet I let me in doubt how those could derived from a limited sample of practitioners, who we do not know about their origins and who they represent.

    a. We have added more information about the respondents and what they represent as well as their opinions (see also comment by other reviewer)

3) The traditional relationship and exchange culture between farmers and pastoralists is not described.

     a.  It is correct that we did not go much into this. While we certainly acknowledge its importance, we find that more elaborate discussions would be repeating what is well known from the extensive ethnographic and anthropological literature.

4) Concern with the methodology in the paper and the number of respondents:

     a.  They paper is basically an exploratory exercise that emanated from the fact that this workshop was held and provided the opportunity to gauge opinions from different stakeholders involved in dissemination of information. We certainly appreciate the limitations of this approach and therefore have also used our experience from field work reported in other papers (Rasmussen et al. 2014 and 2015) as well as the wider literature. We have added some methodological reflections in the conclusion as this topic definitely requires more substantial field based research across a range of different stakeholders.

5) The fine time and spatial scale information requirement is generally a true need for communities but recent CCAFs experience in Senegal (using rural radios), or METAGRI project in Cote d'Ivoire, clearly demonstrated a general interest in aggregated climate services. The sowing dates and length of the growing season were useful for decisions to be made on-farm to adapt to likely seasonal profiles

     a.  We take note of this, but do not really see which changes this should generate in the paper.

6) I was unclear about the merged approach of identifying information needs for farmers and pastoralist. To me the two groups have different information needs but the analysis seems to put them in the same group when exploring data and information requirements.

     a.  It is not our intention to say that farmers and pastoralists have the same needs – but they can use the same types of information for their specific (and often different) needs. We hope this comes across in the paper.

7) The study is more like an expert knowledge assessment and will be easily dismantled by control of facts from practitioners on the ground. The authors recommended a real time system of disaggregated data at a daily basis to help farmers and pastoralist. Is there any example of such a system in the world? Is that possible given the level of technology we have in Africa?

     a.  The study is actually based on statements from practitioners who work on the ground, so we do not agree that it is not rooted in actual practices. There are many examples of real-time information services in the World and the technology is not complicated. In Africa, where the use of mobile technologies are often more advanced than elsewhere, a combination of political commitment and private investments would of course be needed to implement it. We have made these issues more clear in section 3.2.

8) The qualitative information might not be appropriate to generate quantitative evidence and the sample size to me is very limited. The experts might not have covered the range of opinions needed to develop the outputs. The traditional knowledge is not fully accounted in the modern assessment of information needed for informed decisions by herders

     a.  We are well aware that this is not a quantitative and representative sample and although we show a distribution of answers, this is not a quantitative analysis as such, but merely to provide the reader with an overview of responses. The study could, in principle, be replicated by asking the same respondents the same questions (respondents' names are on file, but they are kept anonymous for publication purposes). The figure caption has been

changes to avoid signaling a strict quantitative approach, but rather an overview of responses. See also our response to the previous comment.

9) Also, a distinction needs to be made between data and information. Information adds more value to data and gives examples of application rather than pure data dissemination.

   a. This is a valid point – the paper has been checked and we use the term information now.

10) But the disregard to climate change trends is quite worrisome to me. Many previous studies showed the importance of climate change in the Sahel including perception analysis of such trends. The authors themselves are part of that community.

    a. First of all, we do mention certain elements that might be linked to climate change, e.g. the crucial nature of August rainfall. Secondly, we have to be very cautious when discussing climate change in the Sahel compared to the inherent climate variability. There is no doubt that the droughts in the 1970s and 1980s represented a (temporary?) shift in climate patterns, but this has been difficult to link to anthropogenically driven climate change. That climate change will have an influence in the future is certain, but for now models at regional scale have great difficulties in agreeing on predictions of climate change for the region. Given the limited number of meteorological stations in the region, it is also very difficult to claim actual changing climate in the past. Hence, we agree with the referees that acknowledge the highly variable climate in the Sahel as a fundamental condition in which resource conflicts exist. In previous work that the referee alludes to, we have shown that climate factors actually have a limited effect on farming and pastoral decisions, probably because the variable climate has always existed. As mentioned above, we have rephrased to use 'weather' rather than climate in the analytical part of the paper as this is what the information systems we deal with are about.

11) Figure 1. We do not know is that is % or absolute values. In any case I doubt about representativeness of the information given the very small sample size?

    a. The pie chart indicates absolute numbers – this is now indicated in the graph. The question of representativeness has been answered above.

Anonymous referee #4:

Thank you very much for a very detailed review and the positive notes on the relevance. We have responded to all of the constructive suggestions, most of which we can address:

1) Initial statements on pastoralist-farmer conflicts need to be better connected to the peer-reviewed literature (e.g. the first two references are working papers).

   a. It is not entirely correct that they are working papers – they are published as a book chapter and in a series. But point taken - additional peer-reviewed literature has be cited here

2) References in the introduction are quite selective (e.g. only citing three papers on the climate-conflict link, despite a body of recent literature).

   a. Perhaps the referee could point us to some key references. We thought we had captured some of the most relevant for the current debate. As we focus on small-scale resource

conflicts and not the wider climate-conflict links, the latter is mainly used to set the context for the paper.

3) Since the paper aims to enter new ground by connecting different issues, a more systematic review is needed on pastoralist-farmer conflicts related to key resources (land, water, livestock, crops, ..) and why information on those is relevant for farmers and pastoralists in the Sahel, as suggested in the paper.

    a. We acknowledge the benefits of doing systematic reviews, but we find that it is beyond the scope of this paper to do a full systematic review of the pastoralist-farmer conflict literature. We believe that we have captured the (few) papers that discuss information needs in relationship to farmer-pastoral conflict but if we have missed some important pieces, we will of course be pleased to integrate these. The paper is essentially an explorative piece of ideas and to implement a systematic review procedure at this point would not be possible. It probably would also not be worth the effort given the limited literature that speaks of climate and resource information for pastoralists as documented in the sources used in the paper (Rasmussen et al. 2014, 2015).

4) Further it is important to make the meaning of conflict used in this paper more explicit which apparently deals with small-scale conflicts.

    a. Point taken – this has been corrected in the introduction (p.3

5) Some statements deserve better justification, e.g. the authors associate references with "simplistic explanations" but it is not discussed why they are simplistic and which better explanations could be given

    a. Agree, the term 'simplistic has been removed as the diversity of explanations is mentioned in the following sentences, so no need to say this.

6) Drawing on the literature more specific research questions and hypotheses could be derived, leading to the core part of the paper, the discussion of different categories of information sources, means of communication and relevant data

    a. We are not entirely sure what is meant here - we believe that we are indeed doing what is suggested here.

7) Comments related to the survey at the workshop and the figure.

    a. We realize that figure 1 does indeed appear as an attempt to make a quantitative assessment of the data, but in reality it was meant to give the reader an overview of the responses. We have revised the caption to make it clear that it is small sample and that the figure is mainly to provide an overview.

    b. The figure represents absolute numbers

    c. We have reported most of the statements of interest from the questionnaire and discussions.

8) No theoretical explanations are given whether and when information is leading to competition, sharing or better distribution of resources. Did conflicts emerge because of correct information or due to wrong and lacking of information? This could be identified as a research questions earlier in the paper

    a. We do have an example of conflict arising because of 'wrong' information – to quote the respondents. The answer was not elaborated on by the respondents and it could be an issue of how the information was interpreted along the communication pathway rather

than actually being wrong information. We have added this example to the analysis, but not developed it into a full research question.

9) The paper emphasizes agreement among the workshop participants "that there is a need to improve both the quality of information and how it is disseminated" (page 8). This raises the questions which indicators could be used to measure improvement and how to improve them
    a. Very relevant point. We do discuss models for improving, but not indicators for measuring their success. We thought it would be somewhat speculative to list indicators, but have included this as a recommendation in the conclusion.

10) Although representatives from three countries (Burkina Faso, Mali and Niger) were participating, almost no country-specific experiences are presented. Burkina Faso was shortly mentioned twice, Mali once and Niger not at all. Any information on the differences or similarities of these countries would be helpful
    a. Partly because of the limited number of respondents, we decided not to discuss country differences too much. However, based on the discussions there was a clear difference not so much between nationality, but more based on whether they worked in more or less pastoral areas. We have added information on this at the end of the introduction, but not as part of the analysis.

11) A new type of conflicts between farmers and institutions from information dissemination (page 7). This is an interesting point that could be elaborated further.
    a. Yes, we have added some more comments on this in the analysis as it is indeed a type of conflict that may arise more if information systems are developed and implemented to a higher extent in the future

12) The paper emphasizes how important traditional ways of information and communication are and to ensure the participation of pastoralists (page 8). Here it would be valuable to include a little more about these traditional ways.
    a. These traditions have been documented in detail by Rasmussen et al. (2014, 2015), and details are already provided in section 2.

13) Generally it would be useful to have a table on conflicting issues and how information could address them, following the classification mentioned in C). This might include cases where similar information among different groups leads to similar strategies that increase the risk of overuse and depletion of resources, as well as cases where more options are created that reduce conflict and the added value of modern vs. traditional information dissemination.
    a. Good idea, but we are not entirely sure how to construct such a table without making some unfounded simplifications. Hence, we decided not to do this.

14) Another point that deserves further discussion is the empowerment of pastoralists by equitable access to information to influence land use policies (page 9). How are empowerment and information related in this context?
    a. This was also raised by another referee and we have added some elements to this in the discussion.

15) The conclusions are too short and unspecific to represent the content of the paper.
    a. Point taken – conclusions have been elaborated (but still kept relatively short).